# Comparative Analysis of Milk Microbiomes and Their Association with Bovine Mastitis in Two Farms in Central Russia

**DOI:** 10.3390/ani11051401

**Published:** 2021-05-14

**Authors:** Sergei Sokolov, Ksenia Fursova, Irina Shulcheva, Daria Nikanova, Olga Artyemieva, Evgenia Kolodina, Anatoly Sorokin, Timur Dzhelyadin, Margarita Shchannikova, Anna Shepelyakovskaya, Natalia Zinovieva, Fedor Brovko

**Affiliations:** 1Laboratory of Microbiology, L.K. Ernst Federal Science Center for Animal Husbandry, Dubrovitsy 142132, Russia; dap2189@gmail.com (D.N.); vijmikrob@mail.ru (O.A.); Kolodin77@mail.ru (E.K.); n_zinovieva@mail.ru (N.Z.); brovko@bibch.ru (F.B.); 2Laboratory of Immunochemistry, Shemyakin and Ovchinnikov Institute of Bioorganic Chemistry of the Russian Academy of Sciences, Pushchino 142290, Russia; phursova_k@rambler.ru (K.F.); loskutova-i@mail.ru (I.S.); mshannikova@gmail.com (M.S.); shepelyakovskaya@rambler.ru (A.S.); 3Laboratory of Plasmid Biology, Federal Research Center “Pushchino Scientific Center for Biological Researches”, G.K. Skryabin Institute of Biochemistry & Physiology of Microorganisms of the Russian Academy of Sciences, Pushchino 142290, Russia; 4Laboratory of Cell Genome Functioning Mechanisms, Federal Research Center “Pushchino Scientific Center for Biological Researches”, Institute of Cell Biophysics of the Russian Academy of Sciences, Pushchino 142290, Russia; lptolik@gmail.com (A.S.); dzhelyadin@bibch.ru (T.D.)

**Keywords:** bovine mastitis, milk microbiome, microbial diversity, *Staphylococcus*, *Aerococcus*, Central Russia

## Abstract

**Simple Summary:**

Bovine mastitis is one of the most common diseases in cattle farms in Russia. In this work we investigate a microbial composition of milk samples collected from farms of Russian Central Region. Our data revealed significant dominance of several operational taxonomic units corresponding mostly to groups of *Staphylococcus aureus*, *Aerococcus* spp. and *Streptococcus* spp. We identify interesting fact of Staphylococcus and *Aerococcus* genera seemed to be antagonistic to each other, and the disappearance of the *Aerococcus* genus in milk microbiota may represent a marker for its transition to a subclinical stage of mastitis.

**Abstract:**

Bovine mastitis is a widespread infectious disease. In addition to the economic damages associated with reduced milk yield due to mastitis, the problem of food contamination by microorganism metabolites, in particular toxins, is also a concern. Horizontal transfer of microorganisms from animal populations to humans can also be complicated by antibiotic resistance. Therefore, bovine mastitis is relevant to the study of microbiology and veterinary medicine. In this study, we investigated the microbiome of milk samples from healthy cows and cows with different forms of mastitis from individual quarters of the udder of cows during first and second lactation. Total DNA was extracted from milk samples. The V3–V4 regions of the bacterial 16S rRNA genes from each sample were amplified to generate a library via high-throughput sequencing. We revealed significant dominance of several operational taxonomic units (OTUs) corresponding mostly to groups of *Staphylococcus aureus*, *Aerococcus* spp., and *Streptococcus* spp. In addition, we unexpectedly identified *Streptococcus thermophilus* in samples with high SCC quantities. We found some infectious agents that characterized summer mastitis. We demonstrated that in Central Russia, mastitis is associated with a wide variety of causal organisms. We observed some differences in the diversity of the two investigated farms. However, we did not find any significant difference among healthy, mastitis and subclinical samples according to their SCC status from either farms by principal component analysis. Linear discriminant analysis effect size (LEfSe) confirmed the presence of several indicator genera in farms from Moscow and the Tula Region. These results confirm the complex bacterial etiology of bovine mastitis.

## 1. Introduction

Bovine mastitis is one of the most economically important diseases in cattle [1]. Mastitis is an inflammatory disease of the mammary glands and similar to all other host-pathogen related diseases, has its own signature of microbiota composition [2]. Several studies has demonstrated that it is not possible to exclude transmission from animal to human [3,4,5,6,7], and dissemination of milk of pathogenic microorganisms may colonize both animals and humans, causing secondary diseases [8,9]. Another important aspect of this disease is the probable presence of toxins in milk produced by the pathogens, as many bacterial toxins are known to facilitate allergy development [10,11,12,13]. A number of studies have attempted to identify the primary cause of mastitis and the major microbial pathogen by analysis of microbiota from healthy animal milk compared to milk from animals with various forms and stages of mastitis [14,15,16,17,18,19,20,21]. Further, mastitis may depend on the season, so summer mastitis may be associated with microbiota transmitted from insects. In winter, mastitis can be caused by hypothermia and use of various mats. Injuries of the udder are another important factor in mastitis [22,23,24].

Interactions between multicellular host organisms and microbiota that colonize various parts of the host organism are currently an actively investigated topic. There is a quest for symbiotic microorganisms that are able to provide protection from various pathogens. Classic culture methods have shown a connection between particular bacterial species and the onset of mastitis. Recently developed omics methods are currently in use for expanding the list of such species and identifying their physiological state [25,26].

The aim of this study was to quantitatively analyze the milk microbiota from healthy animals and animals with various forms and stages of mastitis in farms of Central Russia. In the present work, we investigated bovine milk microbiomes on two farms in different regions (Moscow and the Tula Regions).

## 2. Materials and Methods

### 2.1. Sample Collection

Milk samples were collected from Holsteinized black-and-white breed cows from two farms (Moscow and Tula regions) of Central Russia from June to August. Samples were taken 60-70 days after calving from each quarter of the udder separately. Examination of the general condition of the animal by using veterinary medicine methods the mammary gland and inguinal lymph nodes was carried out. To diagnose latent mastitis on a farm, tests of milk samples with dimastin were conducted. Cows had first or second lactation, animals of the same age and lactation were selected. Samples were collected after milking into sterile tubes in compliance with aseptic rules. Classical hygienic procedures included cleaning of teats with an individual paper towel before milking and post milking teat dipping in 0.5% solution of chloramine, after which the udder was dried with an individual paper napkin. All samples were transported to the laboratory under refrigeration (4–8 °C) in cool boxes with ice packs. Analysis of somatic cells was performed by the express method on a Somacount 150 (Bentley Instruments Inc., Chaska, MN, USA) analyzer. According to the veterinary data and the number of quarter somatic cell count (QSCC), all samples were divided into three groups of mastitis form: healthy (H) with QSCC up to 100,000/cm^3^; mastitic (M) with QSCC over 500,000/cm^3^, and subclinical (S) the remaining samples. In order to select the most healthy samples we specially lowered the SCC threshold to 100,000. Salt meat broth (HiMedia Laboratories Pvt., Ltd., Mumbai, India) was inoculated with the milk samples at a ratio of 1:9 and stored at 37 °C for 18–24 h. Internationally recognized traditional phenotypic methods, such as Gram-stained colony microscopy, growth in Baird–Parker agar (HiMedia Laboratories Pvt., Ltd.), hemolysis on azide blood agar Pronadisa (Condalab, Madrid, Spain), plasma coagulation, and biochemical identification, were applied to all the isolates. Therefore, 53 milk samples from 31 cows were ultimately used for this study: 18 samples (8 from Moscow rand 10 from Tula regions) derived from clinically healthy, culture negative quarters with an SCC of less than 100,000 cells/cm^3^ (group 1, (H) healthy); 24 samples (13 from Moscow and 11 from Tula regions) derived from clinical mastitis quarters (group 2, (M) clinical); and 11 samples (5 from Moscow and 6 from Tula regions) derived from clinically healthy quarters with an SCC 100,000–500,000 cells/cm^3^ (group 3, (S) subclinical). Additional data about milk samples descriptive characteristics can be found in Appendix A.

### 2.2. DNA Extraction and Sequence Library Preparation

Milk aliquots from Tula and MR farms were kept frozen at −70 °C until DNA isolation procedure. The extraction of DNA from milk samples of both datasets was performed by the same researchers’ team in the same laboratory in a laminar flow cabinet using Milk Bacterial DNA Isolation Kit (Norgen Biotek, Thorold, ON, Canada) according to the manufacturer’s instructions. DNA was subsequently used as a template for polymerase chain reaction. The V3–V4 regions of the bacterial 16S rRNA gene from each sample was amplified using a universal primer set (16S Amplicon PCR Forward Primer = 5′-CCTACGGGNGGCWGCAG; 16S Amplicon PCR Reverse Primer = 5′-GACTACHVGGGTATCTAATCC) with corresponding Illumina (Illumina Inc., San Diego, CA, USA) overhang adapters. The KAPA HiFi HotStart DNA Polymerase was used for the DNA amplification. After quantification and purification of the PCR products, a sequencing library was generated using the 16S Metagenomic Sequencing Library Preparation protocol. Finally, the library was sequenced using a MiSeq 2 × 250.

### 2.3. Bioinformatics Analysis

Generated sequences were processed using the Nephele platform [27], and sequence quality was confirmed with the FastQC toolkit (Babraham Bioinformatics, Cambridge, UK) [28]. The #6494pp sample did not pass quality control and was excluded from further manipulations. Operational taxonomic units (OTUs) were characterized by matching to the SILVA dataset of 16S rRNA sequences [29]. Reads were subsequently mapped back to OTUs to determine OTU abundance for each sample, and differences in the community composition/structure in each sample were analyzed with the Quantitative Insights Into Microbial Ecology (QIIME) package [30]. 

Diversity was assessed by calculating the Shannon diversity metric, the Chao1 estimate of diversity, and the number of observed species for each sample at various sequencing depths.

Beta diversity estimates were calculated using weighted and unweighted UniFrac distances between samples [31].

Linear discriminant analysis (LDA) effect size (LEfSe) [32] available at https://huttenhower.sph.harvard.edu/galaxy (date accessed 1 March 2021) was used to assess microbial compositional differences between the two groups of samples at the genus or higher taxonomic level.

## 3. Results

### 3.1. Taxonomic Profile

To profile the milk microbiome, pools of 16S amplicons were processed to remove low-quality reads, chimeras, and samples with a low depth of coverage. The resultant dataset of 2,870,927 high-quality reads (median reads per sample = 60,413) was rarified to an even depth of 10,885 reads and binned into OTUs at 97% sequence identity. We observed OTUs from 16 bacterial phyla with an average relative abundance of above 0.1%, and six of these phyla (Firmicutes, Proteobacteria, Actinobacteria, Bacteroidetes, Tenericutes, and Fusobacteria) collectively accounted for more than 97% of all sequencing reads (Figure 1). 

Core microbiome analysis identified most abundant bacterial genera, including *Staphylococcus*, *Aerococcus*, *Streptococcus*, *Enterobacter*, *Macrococcus*, *Corynebacterium*, *Acinetobacter*, *Psychrobacter*, *Ignavigranum*, and *Atopostipes* (Figure 2). Many of these dominant core microbiome genera belonged to Firmicutes (especially *Bacilli*), as well as to Proteobacteria and Actinobacteria.

All milk samples analyzed here revealed great microbial diversity, regardless of their SCC status. In our samples, *Staphylococcus* spp. and *Streptococcus* spp. were among the most prevalent genera in all groups. The clinical mastitis group included animals with decreased appetite, depression of the general condition, if at least one quarter of the udder is swollen, affected, enlarged, has compaction, there are clots during milking and pus admixtures, also a positive test with dimastin (analogue of California mastitis test).

In particular, *S. aureus* was found in all three groups examined. In M samples, its highest abundance varied from 41.19 to 95.82%, while in S samples it did not typically exceed 8.92% (in one sample (#7478lp) it was 20.88%), and in H samples with predominating *S. aureus*, these values were in the range of 21.0 to 91.15%.

Sequences belonging to the *Streptococcus* genus were abundant only in samples derived from quarters that had an SCC greater than 430,000 cells/mL, and *S. pyogenes* (90.88%) was likely associated with clinical mastitis in at least one case (#7029pp). Interestingly, the nonpathogenic *S. thermophilus* is not a causative agent for infection, but this species was detected in large amounts (up to 96.37%) only in milk samples derived from quarters with clinical mastitis. 

*Acinetobacter* sp. was detected in most subclinical and clinical mastitis milk samples, except for those with predominating *Staphylococcus* spp. and *Streptococcus* spp. Other possible causative agents for subclinical mastitis could be *Streptococcus* spp. (93.95%, #7197pl), *Enterobacter* sp. (59.32%, #7029zl), and *Macrococcus* sp. (60.26%, #7284pp). Additionally, one case of clinical mastitis (#7211pp) was presumably associated with the predominance of *Mycoplasma* spp. (40.09%).

Species level analysis indicated that *Aerococcus* spp. was the most prevalent genus in 44% of our healthy milk samples.

Sequences representing *Atopostipes* and *Oligella* species were found in relatively high quantities (up to 7.09% and 32.8%, correspondingly) only in milk samples derived from the Moscow region. These species were almost absent in Tula milk samples, except for one S sample (#7021pp) with both *Atopostipes* (6.2%) and *Oligella* (6.38%) spp. detected.

Corynebacteria were represented in the most milk samples, but notably, larger amounts of *Corynebacterium* (up to 35.76%) were detected in samples derived from Moscow region cows with second lactation, with greater than 68% of them exhibiting >10% *Corynebacterium* sequences.

### 3.2. Diversity Analysis

#### 3.2.1. Alpha Diversity

The alpha diversity of milk microbial communities was assessed by calculating Chao1 and Shannon indices (Figure 3A,B). The Shannon index is a nonparametric diversity index that combines estimates of richness (the total number of OTUs) and evenness (the relative abundance of OTUs). For example, communities with one dominant species have a low index, whereas communities with a more even distribution have a higher index. Chao1 is a nonparametric estimator of the minimum richness (number of OTUs) and is based on the number of rare OTUs (singletons and doublets) within a sample. The median value is represented as the center line of each box, while the lower and upper limits of the box represent the 25th and 75th quantiles, respectively. Error bars extend to the last data point within the ×. 

We analyzed diversity and evenness in three groups (H, M, and S) and in two sites (Moscow and Tula). Both metrics showed differences between H, M, and S groups, as well as groups from Tula and Moscow regions.

#### 3.2.2. Beta Diversity

Homogeneity of multivariate dispersion (PERMDISP2) was run using Bray–Curtis distances to assess if differences in heterogeneity existed among the samples based on different factors. As shown in Figure 4A, there was no clear separation of milk samples from groups within the plots. The distribution of milk samples from group M revealed an overlap with groups H and S.

However, clustering of the milk samples based on their microbiota made it possible to separate samples from Tula and Moscow regions (Figure 4B).

### 3.3. Discriminant Analysis

Rarefaction curves for each group of samples and for a cutoff value of 0.03 are presented in Appendix A. For LEfSe analysis, we took sequences with a depth > 50,000 reads and number of observed OTUs > 1000 to avoid the very strong influence of sequences where the minority of numbered OTUs are represented by the ultimate majority of total reads (Figure 5).

## 4. Discussion

Investigating the problem of mastitis and searching for causes of its development remain relevant today. Omics technologies have been actively used in attempts to find a solution to this problem. [33]. Understanding the causes of mastitis and its pathogens will facilitate creation of preventive measures to reduce the incidence.

In the current work, we analyzed milk microbiotas associated with bovine mastitis in Central Russia. Milk samples were provided from two farms located in two neighboring regions of Central Russia but spaced approximately 200 km apart. We investigated milk microbiomes of individual quarters of mammalian gland isolated from two farms in during the summer. 

We identified most abundant bacterial genera, including *Staphylococcus*, *Aerococcus*, *Streptococcus*, *Enterobacter*, *Macrococcus*, *Corynebacterium*, *Acinetobacter*, *Psychrobacter*, *Ignavigranum*, and *Atopostipes*.

Determination of the most abundant OTUs shown that *S. aureus* was detected in all types of milk samples (H, M, and S). *S. aureus* is a major intramammary pathogen associated with mastitis, along with *E. coli* and *Streptococcus uberis* [15,19,34,35,36,37]. In previous work, we demonstrated that an acute form of bovine mastitis in Russia during first and second lactations is often characterized by the presence of *Staphylococcus* species in the milk and *Staphylococcus aureus* in particular [38]. Moreover, in the current study, we detected the presence of *Staphylococcus aureus* in huge quantities (21.0 to 91.15%) in five samples from healthy quarters with QSCC < 100,000 cells/cm^3^. We explain this observation by these conditionally healthy quarters actually belonging to animals with diagnosed clinical mastitis. On the other hand, there are milk samples from mastitis quarters without evident predominating species that could serve as causative agents. These results are consistent with the suggestion that mastitis has a distinct etiology and that dysbacteriosis may be one of the causes of this disease [39]. 

In addition in this work we specially lowered the SCC level for healthy quarters to 100,000 to analyze healthy samples. In previous works authors suggest that SCC level varies depending on the mastitis pathogen [40]. In this direction, Grispoldi et al. in 2019 [41] discovered the prevalence of *S. aureus* ranged from 40.74% in milk samples with SCC < 100,000. In addition Oikonomou et al. in 2014 [35] has demonstrated that *Staphylococcus* spp. and *Streptococcus* spp. were among the most prevalent genera in all groups with high and low SCC levels.

In this work we pay the special attention to the SCC level < 100,000 to analyze healthy samples and conditionally healthy samples obtained from cows with higher SCC levels in other quarter. As of today the low level of SCC values remains under discussion, different authors consider different values [42].

Interestingly, *Staphylococcus* and *Aerococcus* genera seemed to be antagonistic to each other, and the disappearance of the *Aerococcus* genus in milk microbiota may represent a marker for its transition to a subclinical stage of mastitis. The replacement of *Aerococcus* with opportunistic microorganisms is clearly visible when comparing healthy milk samples to subclinical and clinical samples. 

The next agent identified in this work was *Corynebacterium*. In recent studies by Porcellato et al. [43] strong negative correlation of *Corynebacteriaceae* and *Staphylococcacea* was detected. In contrast, early positive correlations with *Staphylococcus* were shown for this genus [44]. Here, we did not identify a strong correlation, but most samples contained sequences of *Corynebacterium*. Similar results on the association of *Corynebacterium* with mastitis were reported in previous studies [34,45].

Medically relevant members of the genus *Acinetobacter* are being increasingly recognized as a serious health hazard worldwide. In particular, *Acinetobacter baumannii*, a member of the ESKAPE group of pathogens (*Enterococcus faecium*, *Staphylococcus aureus*, *Klebsiella pneumoniae*, *Acinetobacter baumannii*, *Pseudomonas aeruginosa*, and *Enterobacter* species), is a leading cause of nosocomial infections [46]. In our study, DNA from *Acinetobacter* sp. was detected in 82% of subclinical and 74% of mastitic milk samples, except for those with predominating *Staphylococcus* spp. and *Streptococcus* spp. However, Pang et al. [45] have shown that *Acinetobacter* is more prevalent in healthy group samples. 

*Streptococcus* was the second dominant group in our work. This genus, including *S. uberis*, *S. agalactiae* and *S. dysgalactiae*, is a typical mastitis associated pathogen [35,45,47]. However, in our work, we identified nonpathogenic *S. thermophilus* as a dominant organism in three milk samples with high SCC level (>2,000,000). We speculate that this is due to the ability of increasing somatic cells to stimulate growth of *S. thermophilus* [48].

The possible role of anaerobic bacteria *Fusobacterium* spp. and *Porphyromonas* spp. as opportunistic pathogens in existing mastitis cases was discussed in previous studies [35,49]. In addition, several authors have shown that *F. necrophorum* may be associated with summer mastitis [50,51], which is in agreement with our period of milk sampling (from June to August). Recently, Gu and coauthors have shown that *Fusobacterium* causes dysbiosis [52]. We show here that the prevalence of these microorganisms (17.49% and 16.31%, respectively) may be associated with subclinical mastitis in sample #5895zp.

Moreover, one case of clinical mastitis (#7211pp) was presumably associated with the predominance of *Mycoplasma* sp. (40.09%). Different species of *Mycoplasma* were determined to be mastitis-caused organisms, and *M. bovis* was the most prevalent. *Mycoplasma* sp. stimulates apoptosis and inflammatory cytokine reactions [53] and is one of the reasons for milking reductions [54]. 

Bovine milk microbiome analysis revealed strong variations between quarters, with some quarters clearly dominated by one taxonomic unit, whereas others displayed a more balanced profile. Similarly, a considerable difference in quarter microbiota from the same cow was described in previous work [43] We show in our study that not only a single group of pathogenic microorganisms but several pathogenic genera and species within milk microbiota might be associated with subclinical and clinical bovine mastitis at the same time within one general area of Central Russia.

Analyzing the microbial diversity of Moscow and Tula regions, we determined that diversity of the Moscow region groups was higher than in the Tula groups. 

Analyzing the diversity of H, M and S groups, we detected no significant difference between the Shannon indices of groups M and S. However, the Shannon index of group H was lower compared to previous studies reporting the richness and diversity of microbiota from healthy quarters as significantly higher compared to subclinical and clinical samples [35,45].

We found some differences in OTU with respect to diversity between the farms. *Atopostipies* and *Oligella* genera were detected only in the Moscow farm (except for one S sample). Moreover, when comparing the two farms, *Corynebacterum* was detected in large amounts in the Moscow farm and was associated with second lactation. As shown, early *Atopostipies* was identified in both mastitis and healthy groups, whereas *Corynebacterium* was the most abundant genera in mastitic samples [45]. Moreover, Qadri et al. demonstrated that *C. piogenes* is one causative organism for summer mastitis [55].

Despite the fact that Shannon and Chao1 indices for Moscow region milk samples were higher than in milk samples from Tula, cooccurrence analysis revealed increase marker families in Tula milk microbiomes in contrast to those from the Moscow region. *Bacillales*, *Enterobacteriaceae* and *Leptotrichiaceae* were more prevalent in H Tula samples, while *Flavobacteriales*, *Planctomycetales*, *Pseudomonadales,* and one group of unclassified bacteria were more prevalent in M Tula samples. In the Moscow region, milk samples LEfSe analysis revealed only one significant negative correlation between *Xantomonadales* in the H group and *Neisseriales* in the M group.

## 5. Conclusions

We observed marked difference in certain dominant OTUs in milk samples that have never been reported in related studies. Remarkably, alpha-diversity analysis revealed that the richness and diversity of mastitis microbiota were higher than in healthy and subclinical samples in contrast to previous reports. Beta-diversity analysis showed a clear separation of samples from Moscow and Tula region farms; however, healthy, mastitis and subclinical samples were unable to be distinguished from one another. This indicates that infection-causing agents have an uncertain nature and seem to be common in both farms. We did not identify a significant amount of *Escherichia coli* OTU in either the Moscow or Tula region farm, despite *E. coli* being assumed as one of the frequent inflammatory agents in bovine mastitis. In contrast, other common mastitis causative agents, *Staphylococcus* and *Streptococcus* spp., were widespread in both healthy and mastitis milk samples. Notably, *Streptococcus thermophilus* was preferentially detected in milk samples with very high SCC values. The comparison of OTU composition in milk samples revealed that *Aerococcus* and *Acinetobacter* could serve as marker genera for healthy cows from the Moscow region farm and in cows with mastitis from the Tula region farm, respectively. The role that unidentified bacilli play in normal healthy microbiota remains unclear and requires further investigation.

## Figures and Tables

**Figure 1 animals-11-01401-f001:**
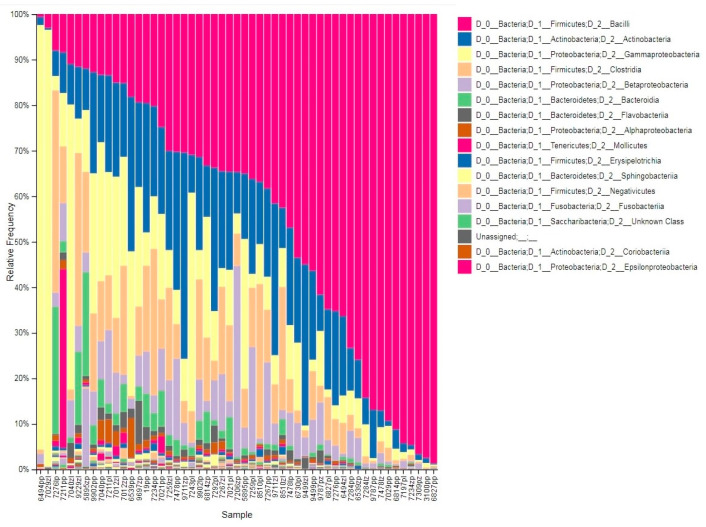
The relative abundance of major bacterial phyla present in milk samples.

**Figure 2 animals-11-01401-f002:**
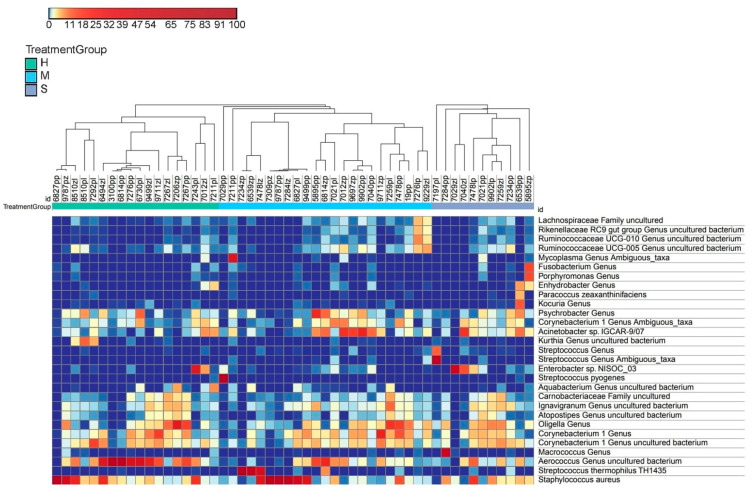
Heatmap of the most abundant OTUs in milk samples. Annotations at the top of the heatmap show clustering of milk samples. The color scale depicts the normalized relative abundance of each OTU. The full OTU heatmap is available as Appendix A. Treatment groups (H, M, and S) are described in Materials and Methods.

**Figure 3 animals-11-01401-f003:**
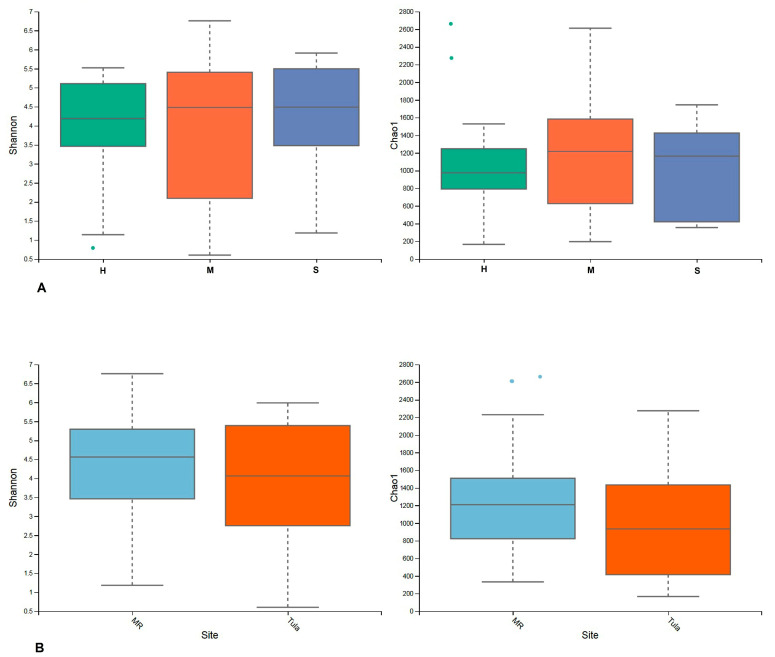
Diversity and evenness. (**A**) Shannon and Chao1 indices for all three groups (H, M, S). (**B**) Shannon and Chao1 indices for both sites (MR—Moscow region, Tula—Tula region).

**Figure 4 animals-11-01401-f004:**
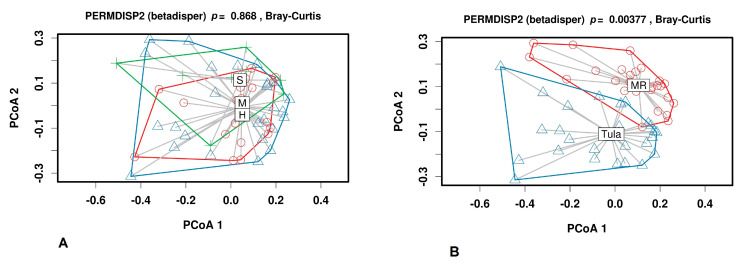
PERMDISP2 beta diversity analysis of microbiota. (**A**) H (red circles), M (blue triangles), and S (green crosses) samples. (**B**) Tula (blue triangles) and Moscow (red circles) Regions.

**Figure 5 animals-11-01401-f005:**
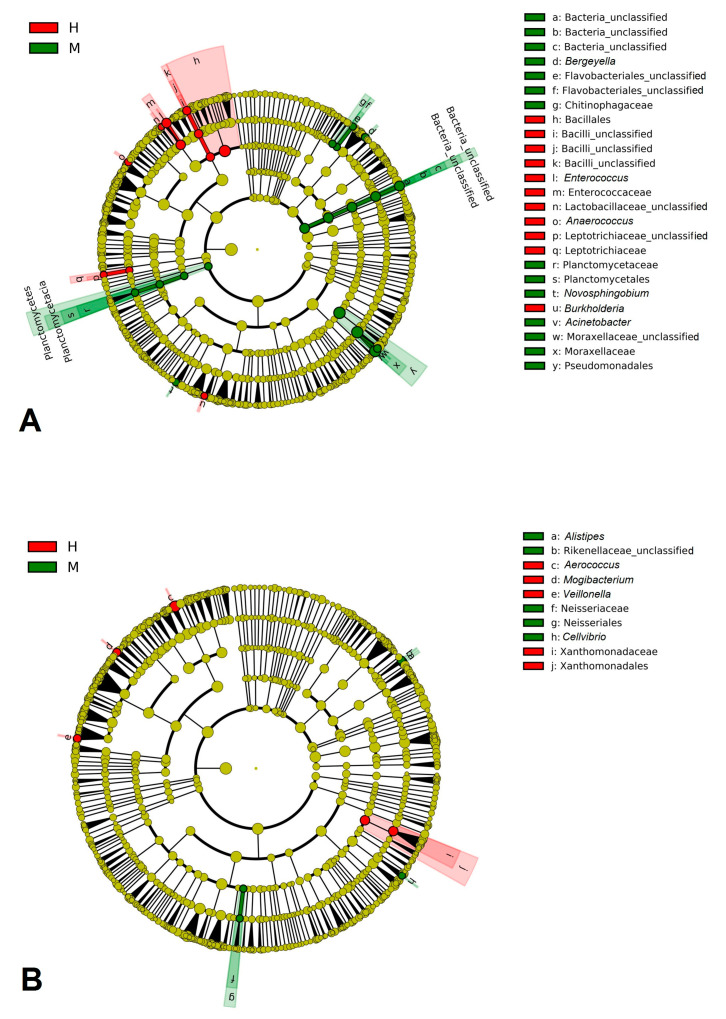
Cladograms based on LEfSe analysis (*p* < 0.05, LDA effect size > 2) of Tula (**A**) and Moscow region (**B**) milk sample microbiomes.

## Data Availability

Sequences generated in this study were deposited in the NCBI sequence read archive (https://www.ncbi.nlm.nih.gov/sra date accessed 11 January 2021) under accession numbers SRX8374271-SRX8374323.

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
