# Peer review of "Comparative Analysis of Milk Microbiomes and Their Association with Bovine Mastitis in Two Farms in Central Russia"

_animals, 2021, doi:10.3390/ani11051401_

Round 1
Reviewer 1 Report
The manuscript presented by the authors is an interesting study on the microbiomes present in milk from dairy cows and their association with clinical and subclinical mastitis. The overall quality of the mauscript is high. I have just two minor suggestion:
- section 2.1, sample collection: do you have the dataset of the characteristics of the cattle? Checking for possible relationship between the microbiomes and age, amount of milk produced etc... could be interesting. How did you choose the subjects for the study? A better description of the experimental design is needed
- discussion: I suggest to check and cite the following papers to improve this section: 1) Grispoldi L, Karama M, Ianni F, La Mantia A, Pucciarini L, Camaioni E, Sardella R, Sechi P, Natalini B, Cenci-Goga BT. The Relationship between S. aureus and Branched-Chain Amino Acids Content in Composite Cow Milk. Animals (Basel). 2019 Nov 16;9(11):981. doi: 10.3390/ani9110981. PMID: 31744129; PMCID: PMC6912583. 2) Grispoldi L, Massetti L, Sechi P, Iulietto MF, Ceccarelli M, Karama M, Popescu PA, Pandolfi F, Cenci-Goga BT. Short communication: Characterization of enterotoxin-producing Staphylococcus aureus isolated from mastitic cows. J Dairy Sci. 2019 Feb;102(2):1059-1065. doi: 10.3168/jds.2018-15373. Epub 2018 Dec 24. PMID: 30591337.
Author Response
Thank you for the very helpful comment with respect to our manuscript. In regard to your comment, this is our reply:
Comment 1: section 2.1, sample collection: do you have the dataset of the characteristics of the cattle? Checking for possible relationship between the microbiomes and age, amount of milk produced etc... could be interesting. How did you choose the subjects for the study? A better description of the experimental design is needed.
Response to reviewer comment no. 1: The authors agree that the studying of relationship between the microbiomes and cattle physiological characteristics could be interesting, but unfortunately, the dataset was limited to characteristics listed in the supplementary table.
Comment 2: discussion: I suggest to check and cite the following papers to improve this section: 1) Grispoldi L, Karama M, Ianni F, La Mantia A, Pucciarini L, Camaioni E, Sardella R, Sechi P, Natalini B, Cenci-Goga BT. The Relationship between S. aureus and Branched-Chain Amino Acids Content in Composite Cow Milk. Animals (Basel). 2019 Nov 16;9(11):981. doi: 10.3390/ani9110981. PMID: 31744129; PMCID: PMC6912583. 2) Grispoldi L, Massetti L, Sechi P, Iulietto MF, Ceccarelli M, Karama M, Popescu PA, Pandolfi F, Cenci-Goga BT. Short communication: Characterization of enterotoxin-producing Staphylococcus aureus isolated from mastitic cows. J Dairy Sci. 2019 Feb;102(2):1059-1065. doi: 10.3168/jds.2018-15373. Epub 2018 Dec 24. PMID: 30591337.
Response to reviewer comment no. 2: The authors are thankful for the provided papers. The reference is added to the list of cited papers.
Reviewer 2 Report
Review ”Comparative analysis of milk microbiome ant their association with bovine mastitis in two farms in Central Russia”. By Sergei Solokov et al.
In short the authors have analysed the microbiota of milk from 53 udder quarters with various amount of somatic cells from two different farms in central Russia. Overall the manuscript is an welcome contribution to the research field of milk microbiota. The abstract summary is in line with the results of the text but certain aspects in the text needs to be clarified.
Materials and methods section:
Sample collection: authors claim that milk samples were collected “in compliance with aseptic rules” and give a hard to follow explanation of the routines for these aseptic rules. This need to clarified so that the reader easily can understand what hygenic measures taken before sampling. Authors claim that samples are taken from each quarter separately but information on number of samples from each cow is lacking, likewise how many cows that are included in the study.
Authors claim that healthy milk samples are culture negative but give no information about bacterial culturing.
Based on information in supplementary material one can guess that authors have chosen cows with high SCC in one quarter and low SCC in an other quarter but this needs to be clarified.
Samples with SCC >500 000 cells per ml are sometimes referred to as mastitic and sometimes as clinical mastitis, a continuous use of words would be desirable.
DNA extraction and Sequence Library Preparation
This section is very rudimentary and impossible for any other researcher to repeat, for instance information about primers and polymerase is missing.
Information about similarities or differences in handling of samples from the two different farms are missing.
Results
Taxonomic profile: Authors mention that sample #7029pp suffer from clinical mastitis, while a definition of clinical mastitis is lacking.
Sequences belonging to Atopostipes and Oligella species are dominant in samples from Moscow region, information if this is caused by a batch effect is lacking.
Beta diversity; The authors are using a PCA-plot to demonstrate lack of differences between H, M and S group, however a multivariate statistical method (such as ANOVA or PERMANOVA) to support this lack of difference would be desirable. The separation of samples from the Tula and Moscow region stress the fact that information about differences on handling samples from the two regions need to be included.
Alpha diversity. In the discussion authors claim that no statistical differences were detected for Shannon diversity, although no information about statistical tests are described in M&M or results. Parts of the text in this section would fit better as a figure caption.
Figures: text, especially for the box plots I very hard to read.
Discussion:
Authors discuss dysbacteriosis as a cause of disease and cite [27] – this is clearly a mistake.
A previously published paper on various SCC and milk microbiota should be cited. Microbiota of cow's milk; distinguishing healthy, sub-clinically and clinically diseased quarters DOI: 10.1371/journal.pone.0085904 . Likewise, previousluy published papers on a core milk microbiota should be cited and included in the discussion. A core microbiota dominates a rich microbial diversity in the bovine udder and may indicate presence of dysbiosis. doi: 10.1038/s41598-020-77054-6
A general comment. Within the research of microbiota and milk microbiota there are publications on biases and effect of contamination. For example: Reagent and laboratory contamination can critically impact sequence-based microbiome analyses. DOI: 10.1186/s12915-014-0087-z . Microbiota data from low biomass milk samples is markedly affected by laboratory and reagent contamination. DOI: 10.1371/journal.pone.0218257 . Although the authors might lack data on the effect of contamination in their data set, a discussion around potential contamination and biases in the data set could be included in the discussion.
Author Response
Thank you very much for the very helpful comments with respect to our manuscript. Below you can find the reply to your comments:
Comment 1: Sample collection: authors claim that milk samples were collected “in compliance with aseptic rules” and give a hard to follow explanation of the routines for these aseptic rules. This need to clarified so that the reader easily can understand what hygenic measures taken before sampling. Authors claim that samples are taken from each quarter separately but information on number of samples from each cow is lacking, likewise how many cows that are included in the study.
Response to reviewer comment no. 1. Classical hygienic procedures included cleaning of teats with an individual paper towel before milking and post milking teat dipping in 0.5% solution of chloramine, after which the udder was dried with an individual paper napkin. 53 milk samples from 31 cows were used for this study. Numbers of samples and cows are listed in a Supplementary table 1.
Comment 2: Authors claim that healthy milk samples are culture negative but give no information about bacterial culturing.
Response to reviewer comment no. 2. Salt meat broth (HiMedia Laboratories Pvt., Ltd, Mumbai, India) was inoculated with the milk samples at a ratio of 1:9 and stored at 37°C for 18–24 h. Internationally recognized traditional phenotypic methods, such as Gram-stained colony microscopy, growth in Baird–Parker agar (HiMedia Laboratories Pvt., Ltd.), hemolysis on azide blood agar Pronadisa (Condalab, Madrid, Spain), plasma coagulation, and biochemical identification, were applied to all the isolates.
Comment 3: Based on information in supplementary material one can guess that authors have chosen cows with high SCC in one quarter and low SCC in an other quarter but this needs to be clarified.
Response to reviewer comment no. 3. In this work we pay the special attention to the SCC level <100,000 to analyze both healthy samples and conditionally healthy samples obtained from cows with higher SCC levels in other quarter.
Comment 4: Samples with SCC >500 000 cells per ml are sometimes referred to as mastitic and sometimes as clinical mastitis, a continuous use of words would be desirable.
Response to reviewer comment no. 4. The usage of such terms has been checked throughout the manuscript.
Comment 5: This section (DNA extraction and Sequence Library Preparation ) is very rudimentary and impossible for any other researcher to repeat, for instance information about primers and polymerase is missing.
Response to reviewer comment no. 5. The V3–V4 regions of the bacterial 16S rRNA gene from each sample was amplified using a universal primer set (16S Amplicon PCR Forward Primer = 5'-CCTACGGGNGGCWGCAG; 16S Amplicon PCR Reverse Primer = 5'-GACTACHVGGGTATCTAATCC) with corresponding Illumina overhang adapters. The KAPA HiFi HotStart DNA Polymerase was used for the DNA amplification.
Comment 6: Information about similarities or differences in handling of samples from the two different farms are missing.
Response to reviewer comment no. 6. Milk aliquots from Tula and MR farms were kept frozen at -70 until DNA isolation procedure. The extraction of DNA from milk samples of both datasets was performed by the same researchers’ team in the same laboratory in a laminar flow cabinet using Milk Bacterial DNA Isolation Kit (Norgen Biotek) according to the manufacturer’s instructions.
Comment 7: Taxonomic profile: Authors mention that sample #7029pp suffer from clinical mastitis, while a definition of clinical mastitis is lacking.
Response to reviewer comment no. 7. The clinical mastitis group included animals with decreased appetite, depression of the general condition, if at least one quarter of the udder is swollen, affected, enlarged, has compaction, there are clots during milking and pus admixtures , also a positive test with dimastin (analogue of California mastitis test).
Comment 8: Sequences belonging to Atopostipes and Oligella species are dominant in samples from Moscow region, information if this is caused by a batch effect is lacking.
Response to reviewer comment no. 8. Since milk samples from both farms were collected by the same team and stored under the same conditions we suppose the absence of the batch effect in this study.
Comment 9: Beta diversity; The authors are using a PCA-plot to demonstrate lack of differences between H, M and S group, however a multivariate statistical method (such as ANOVA or PERMANOVA) to support this lack of difference would be desirable. The separation of samples from the Tula and Moscow region stress the fact that information about differences on handling samples from the two regions need to be included.
Response to reviewer comment no. 9. According to the recommendation, we applied a multivariate statistical method PERMDISP2 for the beta diversity assessment. PERMDISP2 visualizes the distances of each sample to the group centroid in a PCoA and provides a p-value for the significance of the grouping.
Comment 10: Alpha diversity. In the discussion authors claim that no statistical differences were detected for Shannon diversity, although no information about statistical tests are described in M&M or results. Parts of the text in this section would fit better as a figure caption.
Response to reviewer comment no. 10. The Shannon index is a nonparametric diversity index that combines estimates of rich-ness (the total number of OTUs) and evenness (the relative abundance of OTUs). For ex-ample, communities with one dominant species have a low index, whereas communities with a more even distribution have a higher index. Chao1 is a nonparametric estimator of the minimum richness (number of OTUs) and is based on the number of rare OTUs (singletons and doublets) within a sample. The median value is represented as the center line of each box, while the lower and upper limits of the box represent the 25th and 75th quantiles, respectively. Error bars extend to the last data point within the hinge value ± 1.5* the interquartile range. Nevertheless, the authors believe that a figure caption would be better for the indices visual presentation.
Comment 11: Figures: text, especially for the box plots I very hard to read.
Response to reviewer comment no. 11. The authors replaces figures by new ones.
Comment 12: Authors discuss dysbacteriosis as a cause of disease and cite [27] – this is clearly a mistake.
Response to reviewer comment no. 12. The wrong reference was removed.
Comment 13: A previously published paper on various SCC and milk microbiota should be cited. Microbiota of cow's milk; distinguishing healthy, sub-clinically and clinically diseased quarters DOI: 10.1371/journal.pone.0085904 . Likewise, previousluy published papers on a core milk microbiota should be cited and included in the discussion. A core microbiota dominates a rich microbial diversity in the bovine udder and may indicate presence of dysbiosis. doi: 10.1038/s41598-020-77054-6
Response to reviewer comment no. 13. The authors are thankful for the provided papers. The papers are cited and included in the discussion.
Comment 14: A general comment. Within the research of microbiota and milk microbiota there are publications on biases and effect of contamination. For example: Reagent and laboratory contamination can critically impact sequence-based microbiome analyses. DOI: 10.1186/s12915-014-0087-z . Microbiota data from low biomass milk samples is markedly affected by laboratory and reagent contamination. DOI: 10.1371/journal.pone.0218257 . Although the authors might lack data on the effect of contamination in their data set, a discussion around potential contamination and biases in the data set could be included in the discussion.
Response to reviewer comment no. 14. The complete removal of contaminations from the teat or absence of laboratory and reagent contamination cannot be proved, however the authors claims that samples collection was done by the same team on both farms. Collected milk samples were delivered in ice to the laboratory within 2-3 hours. Milk aliquots were immediately frozen at -70 and kept frozen until DNA isolation. The DNA isolation procedure also was performed by the same team at the same laboratory, and at the same time for all the frozen samples. The authors believe that such experimental design allowed if not to avoid but minimize the potential contamination affect.
Round 2
Reviewer 2 Report
The manuscript has improved after the authors review.
I have no further comments on improvements.